# OpenReview forum: "A Disentangled Low-Rank RNN Framework For Uncovering Neural Connectivity and Dynamics"
_ICLR.cc/2026/Conference — ICLR 2026 Conference Withdrawn Submission_

### Official Review · Reviewer_Lyob · 2025-10-28

**Soundness:** 3
**Presentation:** 2
**Contribution:** 2
**Rating:** 4
**Confidence:** 3

**Summary:**

In this paper, the authors propose DisRNN, a method for learning latent representations of neural dynamics that also obey groupwise statistical independence. After assessing the efficacy of their method on a synthetic dataset with known ground truth, they then apply it to two neural datasets, and argue that their method provides more interpretible fits, both in terms of recovered latent dynamics, and in terms of inferred connectivity.

**Strengths:**

- The formulation of low-rank RNNs in terms of a VAE bridges different perspectives and may not be well known.
- The authors not only test their method with in synthetic settings with known ground truth, but also apply it to real neural data.

**Weaknesses:**

- The motivation for the need for such a method was rather weak, and the presentation of the method felt confusing and disorganized at times. For instance, the history convolution kernel used in the formulation of DisRNN seems like an unnecessary complication that is unrelated to the stated goal of the method, and unless I'm mistaken it was also not clarified as to what was actually used for it in experiments (I'm assuming the trivial kernel with the full support at $\tau=0$ was chosen?).
- The partial correlation penalty proposed in equation 10 seems like the real crux of the paper, and that could in principle be applied posthoc with any RNN, including a (noisy) standard low-rank RNN.
- Following on the previous point, for the first experiment, if all methods get approximately similarly good reconstruction of the observation sequences, and latent activities of DisRNN are simply an affine transform of the observation predictions, then surely other methods could yield equally good reconstruction of the "true" latents following an affine transform, no?
- With regards to experiments involving real neural data, the authors seem to conflate interpretible predictions with accurate predictions. For instance, in Fig. 4, even if one accepts the premise that the inferred connectivity predicted by DisRNN is more interpretible (which is certainly up for debate), that is irrelevant if it turns out to be incorrect, and there is no reason a priori to think "separable low-rank channels" is more biologically realistic compared to a "single entangled circuit". Nonetheless, I understand that not having a notion of ground truth here makes it difficult to say more.

**Questions:**

1. With regards to comparisons to other models: what's the difference between LINT and lrRNN in terms of experiment implementation? My understanding is both methods parameterize the weight matrix in a low-rank factorized form.
2. Regarding Fig. 3, I'm not understanding how latent $R^2$ could be affected by rotations of the coordinate system. Rotations are contained within the set of linear transforms, so simply rotating the ground truth trajectories shouldn't affect their linear predictivity. Unless what is happening here is different models are being fit to different rotated versions of the dataset, but the writing seems to imply otherwise.

---

### Official Review · Reviewer_EiA8 · 2025-10-29

**Soundness:** 2
**Presentation:** 2
**Contribution:** 2
**Rating:** 4
**Confidence:** 4

**Summary:**

The paper presents a variational inference framework for fitting low-rank RNNs to neural data. In low-rank networks the $N\times N$ recurrent weight matrix $\mathbf{W}$ can be decomposed by two smaller matrices  $\mathbf{W=AB}$ of size $N\times R$ and $R\times N$. Previous (LINT and SMC; also a VAE) methods fit unconstrained $\mathbf{A}$ and $\mathbf{B}$ matrices and apply a post-hoc orthogonalisation step for visualising the latent dynamics (defined as a projection in the space spanned by the weights). In this submission, an additional regularisation term is applied during training the RNN that promotes a different decomposition of the $\mathbf{W}$ matrix such that the resulting latents are disentangled.

**Strengths:**

S1. The disentanglement term in the loss potentially leads to more interpretable latents compared to previous approaches.

S2. The method shows improvement over baselines on a specific disentanglement $r^2$ metric.

S3. The figures are generally informative and clear.

**Weaknesses:**

W1. The authors write as their first contribution (line 064) that they obtain a generative RNN model. However as far as I understand the authors never show any generation / simulations of the fitted RNN (unlike previous LINT and SMC methods, which did show adequate simulations). In my opinion the author should either adjust their text to reflect they are interested in inference / $p(z|x)$ only, or to also show RNN simulations.

W2. I think the paper requires some clarification of the methods:
 - W2.1 In sections 2 and 3 a history filter was included in the RNN equations. I could not find whether or not this history filter was also applied for the main experiments. If it was indeed applied, I am unable to see how we can disentangle whether the improvements over baseline methods come from the history filter or from the regularisation term. If it was not applied, my suggestion would be to exclude it from the main text (resulting in an easier to read methods section, and less confusion later on), and move the full equations with history filter to the supplementary.

- W2.2. For the inferred latent trajectories, what exactly are you plotting here? In previous works (LINT, SMC and other work on low-rank RNNs) the latents are generally defined as projections of $\mathbf{x}$ on the column space of the weights  / $\mathbf{A}$, here it seems you are (at least for your method) considering latents as projections of $\sigma(\mathbf{x})$on the row-space / $\mathbf{B}$ (equation (4)). Again it would be crucial to verify that the improvements over baselines come from the proposed regularisation, and not from a different definition of how the latents are computed.

- W2.3 There are a couple of other methods related unclear to me, see Question section.

**Questions:**

Q1. What is the difference between lrRNN and LINT, whether or not a Post-Hoc SVD orthogonalisation step was applied to the recurrent weight matrix before computing evaluation metrics?

Q2. The learned recurrent matrices seem very similar between all methods (see e.g., A.2.2) - in fact, if I understand the method correctly, the improvements just comes from how we decompose the recurrent weights? If so could we not also find the desired decomposition easier after training, i.e., find a decomposition of the weights that minimises the disentanglement?

Q3.   You also use your method to fit spiking data, but don't show any plots. Do you indeed learn a good generative model of spiking data when simulating the RNN?

Q4. For the SMC posterior, did you plot the mean of the posterior (with a reasonable number of particles)? The explanation (line 320) for the lower performance in any case seem not right, in the original paper, also fits without using direction labels reached good $r^2$ values - but the authors of that paper seemed to have used a different dataset than here (of a similar task), so we can't make a direct comparison.

---

### Official Review · Reviewer_8SDC · 2025-10-29

**Soundness:** 3
**Presentation:** 3
**Contribution:** 3
**Rating:** 4
**Confidence:** 4

**Summary:**

This paper proposes a method to fit recurrent neural network models that have low-dimensional latent dynamics, which are divided into disentangled groups. Through applications to a few example sets of dynamics, the authors argue that their method outperforms prior approaches, both in terms of accuracy and interpretability.

**Strengths:**

The paper is definitely timely, and the method seems interesting and (up to the issues of uncited work mentioned below) novel. The example applications are mostly convincing.

**Weaknesses:**

The manuscript misses references to closely-related previous work. Most importantly, this is not the first paper to propose a VAE-inspired framework for learning disentangled representations in RNNs. Indeed, there is a previous work by [Miller and colleagues (NeurIPS 2023)](https://proceedings.neurips.cc/paper_files/paper/2023/file/c194ced51c857ec2c1928b02250e0ac8-Paper-Conference.pdf) that in fact adopts the same "DisRNN" acronym! So far as I can see, that prior art goes uncited in this work. See also my **Questions** about other uncited related works.

I also have some more specific questions about benchmarking, which I detail under **Questions**.

**Questions:**

- The authors must compare their method to the existing "DisRNN" method of [Miller et al. (NeurIPS 2023)](https://proceedings.neurips.cc/paper_files/paper/2023/file/c194ced51c857ec2c1928b02250e0ac8-Paper-Conference.pdf). Probably they also should adopt a different acronym to clearly distinguish their method.

- The goal of discovering circuit-level hypothesis (*i.e.*, at the level of parceling the overall observed network into subcircuits) is similar to the goal of the so-called "current-based decomposition" method proposed by [Perich et al.](https://www.biorxiv.org/content/10.1101/2020.12.18.423348v2) in 2020. That work also passes uncited. One limitation - as far as I remember - of the results in Perich et al is that they assume the subnetwork structure is known, which the present method does not. Their method also seeks a post-hoc simplification of a trained, unconstrained network. It'd be interesting to at least comment on the relative merits of these methods.

- I wish the authors did more to explore the effect of the convolution kernel $\psi$. How does the choice of window size $L$ affect the results? In principle, within a linear network one could trade off latent dynamics generated through recurrence with filtering by this kernel. What choices of prior are reasonable here?

- I did not find the choice of synthetic dataset well-motivated. Why did you choose this example? It could be interesting to benchmark your method on a vanilla multi-region RNN, as studied for instance by [Beiran and Clark (2025)](https://www.pnas.org/doi/abs/10.1073/pnas.2404039122).

- It would be helpful if you could provide some investigation of how robust your method is, depending on the rank of dynamics, to subsampling measured neurons within each disentangled group. This is related to the fact experiments will likely be biased in which regions are well-sampled.

---

### Official Review · Reviewer_jvnH · 2025-11-01

**Soundness:** 2
**Presentation:** 2
**Contribution:** 1
**Rating:** 2
**Confidence:** 4

**Summary:**

The manuscript presents a new method to fit recurrent neural networks to experimentally measured neural population responses. Such RNN hold the promise to reveal key features of the mechanisms underlying the responses, both at the level of neural population dynamics and connectivity. The main innovation in the method proposed by the authors is that is identifies solutions for which dynamics factorizes into a small set of “disentangled” components that evolve in separate subspaces of the dynamics. The authors compare their method to several previously proposed methods, both on simulated responses and on experimental data from various neuroscience experiments.

**Strengths:**

The paper addresses a timely question in computational neuroscience

The authors put forward what appears to be a novel approach to fit neural population dynamics.

The proposed methods appear to the technically sound.

**Weaknesses:**

I found the paper quite challenging to parse. The description of the methods and findings in the main text is dense, making an evaluation of the findings challenging.

Overall, the authors could have done much more to demonstrate in what settings and how well their method works in retrieving dynamics and ground-truth parameters, and in what settings and how it fails.

The main simulations used to demonstrate the validity of the approach are based on a ground truth that by design factorizes into disentangled dynamical components. In this setting, it seems plausible that a method that explicitly incorporates such disentangled representations may fare better than alternative methods that do not make this assumption.

Considering the motivation of the work, the authors could have done much more to characterize to what extent various components of their model are identifiable or not. The authors limit themselves to fitting models without inputs, even though in the fits of experimental data such inputs arguably would be required to explain the responses, but would increase issues with non-identifiability.

The premise of the work is also somewhat questionable. In general, why would it be desirable to find solutions that have disentangled dynamics of the kind retrieved by their method? The goal of such RNN fits, as discussed by the authors themselves, it to get insights into the possible mechanisms and connectivity underlying the measured responses. To this end, the ultimate goal should be to retrieve the dynamics, mechanisms, connectivity that most closely resemble the ground truth, not those that result in disentangled representations.

**Questions:**

How well does their method work in cases where the ground truth does not match their hypothesis of disentangled dynamics? Are there cases where their method does “hallucinate” disentangled dynamics (e.g. because it allows a description of the dynamics with overall fewer parameters) when the ground truth is not disentangled? What kind of dynamics are recovered in cases when the ground truth includes external inputs, but their fit does not?

What is the role of the history convolution kernel? Why is it necessary, i.e. could its effect no be effectively mimicked by the recurrent dynamics itself? Does the inclusion of this kernel not result in issues with identifiability of the recurrent dynamics vs. the kernel?

Can the authors independently validate the existence of the inferred subspaces of disentangled dynamics? For example, should the these subspaces not have a reflection in the structure of the covariance matrix of the data?

---

### Note · Authors · 2025-11-21

I have read and agree with the venue's withdrawal policy on behalf of myself and my co-authors.